# Evaluating the Uptake of the Canadian Standards Association (CSA) B701:17 (R2021) Carer-Inclusive and Accommodating Organizations Standard Across Canada

**DOI:** 10.3390/ijerph22060907

**Published:** 2025-06-06

**Authors:** Brooke Chmiel, Allison Williams

**Affiliations:** School of Earth, Environment and Society, McMaster University, Hamilton, ON L8S 4L8, Canada; awill@mcmaster.ca

**Keywords:** carer-employees, caregiving, workplace, care economy, policy, labour, economics

## Abstract

In Canada, 67% of unpaid caregivers are simultaneously balancing paid employment with unpaid care, equating to over 5.2 million Canadian Carer-Employees (CEs). This balancing act often incurs negative impacts on CEs’ health and well-being, including burnout, resulting in adverse effects on their labour force participation. To mitigate these social and economic impacts, McMaster University partnered with the Canadian Standards Association (CSA) to develop the CSA B701:17 (R2021) Carer-inclusive and accommodating organizations standard and accompanying handbook B701-18HB Helping worker-carers in your organization. Since publication in 2017, there has been minimal uptake of the Standard across Canadian workplaces, with just 1062 complimentary downloads total. To determine the level of uptake across workplaces in Canada, the present mixed-methods study used purposive sampling to collect survey (*n* = 71) and semi-structured interview data (*n* = 11). The survey data was analyzed for descriptive statistics and logistic regression modelling. The interview data were thematically analyzed for common CFWPs and barriers to Standard uptake. It was found that only 24% of workplaces have implemented the Standard into their workplace practices, with full implementation and current supports as strong predictors of formal uptake. Prominent themes around barriers to uptake and existing organizational policies highlight the critical importance of workplace culture in facilitating CFWPs.

## 1. Introduction

The economic sustainability of a society depends largely on the readiness and adaptability of its labour and employment markets to meet the needs of a population [1]. The landscape of Canadian labour markets and workforce is changing, impacted by shifts in population demographic trends and cultural values [2,3]. While the aging population in Canada is rapidly increasing as a result of higher life expectancy rates [4], declines in population growth and fertility rates [5] are indicative of significant demographic changes in the coming years. In 2022, older adults (age 65 or older) made up 19% of Canada’s population, with an estimated increase to 23% in 2030 [6]. This evolving landscape presents an increase in the need for care and impacts those providing care. Carer-employees (CEs)—those balancing both unpaid care responsibilities and paid employment—are among those impacted [7]. This includes a range of care responsibilities from doctors’ appointments and transportation to physical and emotional support in the home [8]. In Canada, 67% of unpaid carers are simultaneously employed; this equals over 5.2 million carer-employees [9]. According to Environics Analytics, one in four Canadians will be aged 65 or older by 2043 [10], with an expected increase in the number of CEs as a result. According to Statistics Canada’s 2021 Population Census, the increase in the older adult population places strain on health and home care systems as they struggle to adapt to the influx of Canadians requiring care, especially long-term [11]. As Canada’s population lives longer, labour and employment markets will need to prepare for the resulting increase in CEs as more Canadians age at home, with 51% of care receivers still living at home [9].

CEs experience a variety of challenges associated with the balancing act of unpaid care and paid employment, including negative impacts to their biopsychosocial health resulting from a lack of support in the workplace [12]. Labour and employment markets must adapt to the changing needs of CEs in the workplace to sustainably manage the increase in CEs as the population ages. To mitigate these challenges experienced by CEs, the CSA B701:17 (R2021) Carer-inclusive and accommodating organizations standard (Standard) [13] and accompanying Implementation Guide B701HB-18 Helping worker-carers in your organization (Handbook) [14] was published in 2017, via a partnership between McMaster University and the Canadian Standards Association (CSA). The complimentary Standard and Handbook were designed to provide a framework for organizations to adjust current policies and facilitate a cultural shift towards Carer-Friendly Workplace Practices (CFWPs). Research and evidence highlight the mutual benefits of CFWPs for both employers and employees in maintaining productivity through alleviating strain on CEs with workplace resources and accommodations [7,15,16,17,18]. Although previous intervention research with Canadian workplaces has highlighted the significant health and economic benefits of CFWPs [16,19,20], limited uptake of the CSA B701-17 Standard and Handbook has occurred.

This study evaluates the level of uptake of the Standard across workplaces in Canada. In evaluating the uptake of the Standard, the objectives of the present study are to assess the extent of implementation across workplaces engaging in the Standard, Handbook, and/or CFWPs. Further, this study explores to what extent the organization has implemented a CFWP culture, as well as CFWPs more broadly. The Standard and Handbook have been made available as free downloads for the last 7 years. More specifically, the research question guiding this study is “How are Canadian organizations engaging with the CSA B701 Standard and Handbook, and more broadly, CFWP initiatives outside the Standard and Handbook?”. The CSA retains download data of the Standard and Handbook, including contact information of those who have requested downloads. The assumption is that organization representatives are downloading the Standard and Handbook for use and implementation within their organization, either formally or informally. Once the Standard and Handbook are downloaded, the extent of their use and application are unknown. As noted above, a recent study has shown that many workplaces in Canada continue to lack support for CEs in the workplace [12], despite the existence of the Standard and Handbook. The relevance and importance of Standard adoption/implementation is represented in the ways it provides equitable access to employee accommodations, makes CE supports/accommodations transparent to all employees, acts as a prevention strategy for loss of productivity, turnover, protects psychological health and safety of CEs, and assists in maintaining an inclusive workplace culture with open communication and self-identification of CEs in the workplace. To better understand the contextual barriers to the implementation of CFWPs in Canadian organizations, the present study explores the challenges workplaces experience in establishing CFWPs.

The present study aims to achieve the following objectives through exploration of the research questions: (1) Have organizations that have downloaded the Standard implemented it in the workplace? If so, to what extent has it been implemented? Has the Standard/Handbook been useful in building a CFWP? (2) Has the Standard/Handbook facilitated the creation of CFWP supports within the organization? If so, what supports have been created? How have they been maintained? (3) What are the barriers organizations are experiencing that have downloaded the Standard? (4) What are the barriers experienced in implementing CFWP policies and practices in the workplace? Following a brief literature review, the methodological approach is reviewed before results are presented; a discussion follows the results, including limitations, ending with the conclusion.

The emergence of CEs in the workplace has become more prevalent over the past several decades [21]. Resulting from a variety of factors and shifts in Canada’s economy over the years, including the recent COVID-19 pandemic, CEs in workplaces across the country have continued to increase drastically [22,23]. One in four Canadians provides care to a family member or friend while also balancing paid employment, either full-time or part-time [9]. As the aging population in Canada began to increase while population growth plateaus, the sandwich generation emerged as a prominent demographic in the 1990s, defined as those tasked with providing informal care to aging parents and young children [24,25,26]. Data from Statistics Canada shows the population continues to stagnate in growth due to low fertility rates, while the aging population continues to climb, projecting that the population of older adults will exceed younger adults providing care; older adults are expected to represent up to 25% of Canada’s population by 2036 [11]. Researchers continue to advocate for the economic and mental health benefits of supporting CEs in the workforce for the employer and the employee [17,27,28,29]; the following literature review explores the relevance of CFWPs through the lenses of gender and equity, workplace culture, the COVID-19 pandemic, and demographic shifts.

### 1.1. Gender and Equity

The gendered nature of caregiving continues to persist [30]; 52% of women aged 15 or older make up the majority of formal and informal care in Canada [31]. The Statistics Canada 2018 General Social Survey on Caregiving and Care Receiving found that, as a result of their caregiving responsibilities, women reported negative impacts on their job security and level of productivity more than men [32]. Women spend more time caregiving than men, reportedly spending three hours more per week providing unpaid care [9]. As a result, women are more likely than men to reduce their hours of work, change from full-time to part-time status, take additional unpaid time off work, and leave the workforce altogether [30] while also making on average CAD 20,000 less than men annually [9]. Like many marginalized communities, a lack of awareness around the challenges CEs face both in the workplace and at home often leaves these individuals to suffer in silence without much support to alleviate strain. In 2018, 60% of employees who left paid employment due to caregiving were women [9]. While the government of Canada currently offers financial assistance through the Compassionate Care Benefit (CCB) and Employment Insurance (EI) caregiving benefits, these supports have been deemed inadequate to support the diverse range of caregivers in Canada [33,34,35].

### 1.2. Workplace Culture

Canadian employers have historically been largely unsupportive of CEs in the workplace, despite the economic advantages of providing accommodation to employees to mitigate recruitment and retention [7,15,25,36,37]. Workplaces have an obligation to meet the needs of their CEs as economic and political climates shift. A primary factor contributing to the lack of awareness of the prevalence of CEs stems from challenges around self-identification, often resulting from concerns around stigma and discrimination [38,39]. If CEs do not identify themselves in the workplace, the need for support and accommodation is not recognized. Providing accommodation for CEs has been shown to improve employee retention, satisfaction, recruitment, and productivity [16,18,27,37]. Reciprocally, support for CEs in the workplace results in better overall mental and physical health of CEs, with reduced levels of stress, anxiety, and dissatisfaction [8,12,16,29,40]. As 44% of CEs are aged 45–65 [41], representing the most experienced and specialized employees in the workforce, retention is of relevant concern to employers [42]. Previous intervention research has corroborated the economic and cost-saving benefits of implementing a CFWP from an employer perspective, including reduced absenteeism, turnover, improved productivity, retention, and employee satisfaction [16,18,19,20]. Statistics Canada estimates the annual cost of replacing unpaid care work with employed caregiving is CAD 51.5 billion [43].

### 1.3. COVID-19 Pandemic and Demographic Shifts

The COVID-19 pandemic had major impacts on daily life and economic prosperity, bringing employment and labour markets to a halt for a sustained period of time [44]. Inciting cultural shifts across Canada, values around work–life balance began to change as more Canadians were forced to work from home or in hybrid environments, temporarily be placed on leave, or lose their employment due to financial restraints across workplaces [45,46]. Spending less time at work led many Canadians to reflect on the desired balance between work and personal life, particularly for younger generations. Data published from the Statistics Canada 2021 Census of Population highlights the generational differences in work–life values, with younger generations valuing more of a balance in comparison to older generations. Labour markets must adjust to these changing values [11]. As the population ages and younger caregivers dominate the employment markets [11], employers will need to accommodate the changing values of CEs as they advocate for better work–life balance in providing care for adult dependents.

## 2. Materials and Methods

Recruitment of participants for the present study was conducted in partnership with the Canadian Standards Association (CSA), later expanded to include a broader context of workplaces engaging in CFWPs through an existing organizational network. Within the scope of the present study, the total number of downloads of the Standard and Handbook from the CSA as of 31 March 2024, including both the English and French versions, sits at 1062. Given the minimal downloads of the Standard and Handbook of the French versions, these downloads were excluded from recruitment for the present study. Limitations surrounding the response rate of the target population and subsequent expansion of recruitment methods are outlined in sections below. In addition to recruitment of those who have downloaded the English version of the Standard and/or Handbook, research clearance for this work (MREB #6975) was amended to include the expansion of recruitment through these networks of 30 organizational contacts, as well as students of the “Creating Caregiver-Friendly Workplaces” (CCFWP) course offered through McMaster Continuing Education (MCE). This course was built in partnership with Dr. Williams at McMaster University and MCE, developed on the foundation of the Standard and Handbook. The 10 h complimentary course launched on 2 April 2024 and at the time of recruitment for the present study had 459 students enrolled. The course targets Human Resources (HR) professionals, Occupational Health and Safety (OHS) professionals and employers/managers, offering the option to obtain a microcredential towards professional development https://continuing.mcmaster.ca/programs/health-social-services/creating-caregiver-friendly-workplaces/ (accessed on 1 July 2024). The courses orientation around the Standard and Handbook presented it as an ideal channel of recruitment for the present study—engaging with organizations that are interested in building CFWPs for their CEs. As this study is concerned with a particular demographic, that being organizations that have downloaded the Standard and/or Handbook, as well as engaged with CFWPs more broadly outside the Standard/Handbook, purposive sampling through these channels was determined as the most applicable method for recruitment. Ethical protocols were followed throughout the recruitment process and data collection of both phases of the study. Consent for survey participation was collected on the first page of the survey, and consent for the semi-structured interviews was collected orally and tracked through a consent log using pseudonyms. Respondents of the survey were anonymous unless they chose to provide their contact information to be contacted for a follow-up interview and/or the chance to win an iPad, as part of the strategy to increase the incentive to participate in the study. Interview participants were given the option to review the transcripts for verification and confirmation, allowing them the opportunity to edit any of their responses.

The present study employed a mixed-methods approach with two phases. Participants were contacted through email and invited to participate in filling in the short questionnaire (*n* = 71). Following this, a sub-sample (*n* = 11) of participants was recruited from the questionnaire of those who had opted to participate in a follow-up interview. However, due to survey recruitment limitations bleeding into recruitment of interview participants, over half of the interview participants were recruited having not completed the survey (see Figure 1 below).

The first phase consists of a cross-sectional survey with 33 questions, the majority of which are closed-ended. The questionnaire was concerned with four major content areas; (1) what are the characteristics of the organization (size, sector, access to HR department, position title, number of years at organization), (2) has this organization implemented the Standard/Handbook, (3) if so, whether the Standard and Handbook have facilitated the implementation/creation of CFWPs, supports and accommodations for CEs in the organization, and (4) whether the organization has existing CFWPs outside the formal/informal adoption of the Standard. While initially the questionnaire distribution was restricted to the CSA as the gatekeeper and only point of contact for the target demographic, limitations around efforts in achieving the desired response rate were considered. To mitigate this, the CSA provided a follow-up email at the two-week point following the point of the initial survey distribution. The questionnaire was deployed through LimeSurvey, a survey platform that McMaster University licences. The survey data was analyzed for descriptive statistics and logistic regression modelling. The interviews each lasted between 30 and 60 min, and all but one were conducted through Microsoft Teams. The interviews were recorded and transcribed verbatim. The interviews were thematically analyzed [47] using the Constructivist Grounded Theory (CGT) approach [48,49,50] in Atlas.ti, a form of Computer Assisted Qualitative Data Analysis Software (CAQDAS). This mode of analysis has been identified as the most efficient for qualitative data, supporting the robust identification of themes and fully exploring the data with the aim of satiating the data. Participants were provided the opportunity to verify their responses, explanations, and interpretations, being able to add, edit, retract, and/or validate their responses.

## 3. Results

Findings of the present study are indicative of minimal uptake of the Standard across Canadian organizations within the context of formal adoption of the English Standard into workplace policies and practices. While the survey data provides insight into the landscape of the lack of adoption of the Standard across respondents, the interviews offer more contextual, in-depth analysis into the mechanisms both hindering and helping the adoption of the Standard and CFWPs more broadly within Canadian workplaces. Minimal uptake of the Standard was found to be related to a variety of influential factors established through triangulation of the survey and interview data. Results are outlined below, beginning with descriptive statistics outlining the current uptake of the Standard across Canadian workplaces, followed by regression analysis. Logistic regression modelling suggests organizations that currently offer existing supports for CEs in the workplace (OR = 15.9, *p* = 0.002), and have fully implemented all aspects of the Standard in the workplace (OR = 15.8, *p* = 0.003) are the strongest predictors of whether the Standard is formally adopted into the workplace policies. Qualitative findings explore the facilitators and barriers to uptake of not only the Standard but CFWPs more broadly.

As the survey questions were voluntary, respondents of the survey were not obligated to provide an answer for each question, and consequently, all variables contained missing values. To mitigate this challenge and optimize the dataset for analysis with the use of various methods, imputation was used to fill in the missing observations across the variables. The average of each variable was used to impute the missing observations, with each ranging in the proportion of missing responses for each question. All proportions of blank responses fell under 50% for each variable, the highest being 44% of blank responses in a variable.

### 3.1. Descriptive Statistics

Across the survey respondents (*n* = 71), an overwhelming majority reported from the healthcare ([*n* = 28], 40%) and education ([*n* = 19], 27%) sectors, corroborating the findings from a recent scoping review conducted by Lorenz et al. (2021) [15] which found the healthcare and education sectors to be among the most prominent in availability of CFWPs globally. Many reported an unlisted sector under “Other” ([*n* = 14], 20%), including government, law, non-profit, insurance, and hospitality. Unsurprisingly, the lowest participating sectors represent the primarily male dominated sectors of construction ([*n* = 1], 1.4%), manufacturing ([*n* = 1], 1.4%), engineering ([*n* = 1], 1.4%), and mining/primary resource ([*n* = 2], 2.9%). This suggests further research must be conducted with a focus on increasing the availability of CFWPs across the male-dominated sectors in reducing stigma around male caregivers [51,52,53]. With respect to workplace size, the majority of respondents were evenly split between small (1–99) and large (500+) workplace sizes, sitting at 39% [*n* = 28] for both, while remaining participants were reporting from a medium (100–499) sized workplace ([*n* = 15], 21%). In the interest of better understanding how workplaces are being introduced to the Standard, respondents were asked to identify how they heard about the Standard, the majority of which through email listservs ([*n* = 19], 27%). Notably, this is likely due to the expansion of recruitment, including an email blast to those who have enrolled in the CCFWP course that was built around the Standard, inviting eligible individuals to respond to the survey in response to whether the course has led to the adoption of the Standard in the workplace. Next to email, many participants heard of the Standard through internet searches ([*n* = 12], 17%) and social media promotions ([*n* = 12], 17%), while 16% reported having heard of the Standard through “Other” means, including independent searches on the CSA website. Only 24% [*n* = 17] of respondents reported that the Standard is currently implemented in the workplace, while the majority reported that either it was not currently implemented ([*n* = 28], 39%) or they were unsure ([*n* = 26], 37%) if it was formally adopted as a policy. While a majority indicate the Standard is not formally adopted, many reported the existence of CFWPs already in place to support and accommodate CEs in the workplace, with 42% [*n* = 30] reporting that there are existing workplace supports for CEs outside formal adoption of the Standard in the workplace. Table 1 below outlines the frequency and percentage associated with responses to each survey question.

### 3.2. Logistic Regression Modelling

To better explore the relationship between variables and formal adoption of the Standard by an organization, logistic regression was used to model the relationship between the predictor variables and the probability of these outcomes. Survey questions consisted of various categorical questions, including binary variables “Yes”, “No”, as well as multiple-choice and 5-point Likert scale questions. We created binary variables for “Does your organization have access to an HR department/role”, “Is the Standard implemented at your workplace”, and “Are you or have you been a caregiver”, where the value of 1 indicates the answers of “Yes”. Those with categories “Yes”, “No”, and “Don’t Know”, “Yes” was coded as 1, while “No” and “Don’t Know” were coded as 0. For the workplace sector, given the high ratio of those in the healthcare sector (40% of respondents), the category “Healthcare” was coded as 1, and all other categories in this variable were then coded as 0. For workplace size, small workplaces (1–99 employees) was coded as 1, while medium (100–499) and large (500+) were coded as 0. For the length of time at the organization, 5+ years was coded as 1, while “Less than one year” and “1 to 4 years” were coded as 0. For those identifying as a caregiver, “More than 20 h per week” was coded as 1, with “Less than 10 h per week” and “Between 10–20 h per week” coded as 0. All Likert scale questions were coded as “Strongly Agree” and “Agree” being 1 and “Neither agree nor disagree”, “Disagree”, and “Strongly Disagree” being 0. This was followed by running logistic regression models by considering all possible 20 predictors. Using the step() function in R, backward stepwise selection was used against the model to systemically eliminate the predictors of least value in predicting whether the Standard is formally adopted in the workplace. This function uses the Akaike information criterion (AIC) method to determine which variables have the best fit for the data, starting with all variables in the model and removing the least useful ones one by one until the model with the best fit remains. The step() backwards function runs through the full model, eliminating variables based upon the generated AIC result, where the final model will return the model with the lowest AIC, with the understanding that the model with the lowest AIC explains the greatest amount of variation using the least number of independent variables. This method returned a model with four of the binary variables remaining; according to this model, whether the respondent indicated the Standard was fully implemented in the workplace, the workplace has existing supports for CEs, the workplace values work–life balance and they complete more than 20 h of unpaid care each week were calculated to be the best predictors of whether the Standard was implemented in the workplace. With respect to the strongest and most significant predictors of whether the Standard was implemented, the Odds Ratio (OR) for whether the Standard was reported to be fully implemented in the workplace (OR = 15.8, *p* = 0.003) was the strongest predictor of whether the Standard is formally adopted into the organizations policies, followed by whether the workplace offers existing caregiver supports (OR = 15.9, *p* = 0.002). It should be noted that given the voluntary nature of the Standard, full implementation of the Standard refers to the workplace having implemented all aspects of what the Standard entails; however, formal adoption of the Standard refers to whether the workplace has chosen to recognize the Standard (CSA B701) in formal employment and workplace policy. The variable of whether the workplace values work–life balance suggested importance in determining whether the Standard is implemented, but was less significant (OR = 4.67, *p* = 0.108), suggesting more data is needed for further analysis. Surprisingly, the variable of whether the respondent provides >20 h of care per week (OR = 0.25, *p* = 0.108), although statistically insignificant, suggests that those in this caregiving role are less likely to report formal adoption of the Standard. Table 2 below outlines the OR and CI results of the logistic regression model, including *p*-values for statistical significance.

### 3.3. Thematic Findings

Across a total of 32 codes applied in Atlas.ti, thematic analysis of the interview data presented 4 macro themes across the 11 participant interviews, with a total of 14 micro themes (Table 3). Participant characteristics are outlined in Table 4 below, including sector, organization type, gender, and location.

### 3.4. Workplace Culture

The data make it overwhelmingly clear that support and accommodation for CEs is sustained through the establishment of an inclusive workplace culture that supports CFWPs. The values held by an organization shape the way employees perceive their ability to seek support and accommodation from their employer or organization. The CEO of a small federal non-profit organization, spoke to her reasoning for implementing the Standard within the organizations workplace practices, recognizing the importance of instilling an inclusive culture: “…it’s important to create a culture and an environment where people felt safe to be vulnerable and say, ‘I need’ or ‘I would appreciate’…” (P9, F). Establishing and maintaining an inclusive workplace culture is a central component to effectively adopting the Standard into practice sustainably, recognizing that different workplace sizes and sectors must adapt their policies and procedures to support a broader operational definition of inclusivity, workplace culture must recognize the evolving landscape of an employee’s circumstances and that changes will occur over the life course, often temporarily: “ your situation may be this today, but I want to retain you, and I want your loyalty… because you need to take time to be with a loved one… we’ll work with you to find a solution” (P9, F).

Whether positive or negative, workplace culture was observed to be a major catalyst in whether the Standard was implemented in the workplace or not, as discussed by P4 (F): “…it’s always been a matter of, you know, figuring out how to make it work for our employees and their families and the business… it’s not a problem that needed solving, and our employees have never shown any indication, and they’ve never given us any feedback to say… I can’t balance my life here” (P4, F). As the Senior Advisor of Workplace Experiences at a consultancy company, P4 recognized that the existing positive inclusive culture within the workplace negated any potential need to implement a formal policy or Standard around supporting CEs in the workplace. What the Standard offered was already implemented in the workplace, and therefore, formal adoption of a Standard was deemed to be unaligned with the organization’s values around informal support and accommodation. Alternatively, P3, an Early Childhood Educator with a school board in Canada, discussed a lack of inclusivity considerations within the workplace culture: “You kind of go in and you do your job… But I would say the general consensus is nobody talks about it” (P3, F). The restrictive workplace culture reported by P3 (F) contributes to and shapes the perceptions of employees within the organization of what support they may request and receive from a manager or supervisor, acknowledging that formal adoption of policies is not enough to ensure that CEs receive the support and accommodation needed in the workplace: “Policies are great on paper but implementing them and truly being a part of the workplace… is another thing” (P3, F). Perceived barriers to implementing the Standard as established by P3 (F) reflect the significant importance of instilling a workplace culture that supports employees feeling comfortable communicating their needs and opening the dialogue of communication between employees and employers.

#### 3.4.1. Communication

Facilitating an inclusive workplace culture that allows for the adoption of CFWPs requires honest and open communication between the CE and the employer: “Those are really difficult conversations for many to raise and so you have to have an open relationship with your employer… a trusting relationship with your employer, and many don’t have those opportunities… these are circumstances where you really have to be reliant on opening up those channels of communication…” (P7, F). While the Standard offers the formality of policies in support of CEs within the workplace, if the workplace culture does not encourage open communication between employers/HR and employees, it may hinder CEs from seeking or accessing the supports and accommodations available to them in fear of being reprimanded. Effective communication within an organization between employees and senior management/HR may contribute to the facilitation of whether the Standard is formally or informally adopted, sometimes resulting in the organization selecting certain components of the Standard to put into practice: “… the fact that we kind of had that open and clear communication allows us to not feel like someone’s gonna take advantage of the policies or take advantage of a situation” (P11, M).

While referencing the Standard for structuring of certain workplace procedures and policies, P11’s (M) informal adoption of such methods was observed to be contingent on the workplace culture maintaining openness and understanding in communication between CEs and superiors/HR. Informal adoption of the Standard or select aspects of the Standard was observed to be understood as instilling certain cultural values, supports, and/or accommodations within the workplace’s current practices, without formal acknowledgement of a policy.

#### 3.4.2. Self-Identification

The idea of self-identification as a CE in the workplace is irrefutably intertwined with an inclusive workplace culture. P7 (F) recognized employees feel comfortable reaching out when the workplace culture and organizational values support the understanding that employees have responsibilities outside the workplace that can overlap with the workplace: “…recognition that you are part of a larger community, not just the work community, and that’s your family. And how do you look after your family?”. P9 (F) discussed the relationship between self-identification in the workplace as a caregiver and an inclusive workplace culture: “Are they just so terrified at risking losing their employment to say anything at all?”. Employers will not see a need to implement the Standard without identification of CEs: “It is very much like which comes first, right?… the stigma or the self recognition… people are like oh I don’t need to like fund this part of my benefits for my employees’ programmes because there’s no one” (P8, F). Communication between the employer and employee must recognize the importance of the two-way dialogue in sustaining a workplace culture that supports the implementation of the Standard, as noted by P11 (M). Self-identification is a critical component to customizing employee support and accommodation to better reflect the needs of the employee: “We’ve had people who identify now a little bit more openly. And so, then we’re able to accommodate a little bit better as well” (P10, F). Umbrella policies can set a certain standard for what is available to employees; however, without self-identification, individual needs of an employee can be missed and left unaccommodated.

#### 3.4.3. Employee Accountability

Scepticism around employee use of the Standard acts as a barrier to its uptake: “…there’s this fear that it’s gonna be abused” (P5, F). Although P5 (F), a Senior HR Leader at a federal non-profit organization, reflects on their experiences with employees accessing workplace accommodations generally being fair and honest: “…like in my 20 years of HR, I only had one case where a policy was abused” (P5, F). An inclusive workplace culture often contributes to facilitating employee accountability in accessing CFWPs, whether related to formal adoption of the Standard or CFWPs more broadly, noted by P11 (M): “…and the fact that I you know that we kind of had that open and clear communication allows us to not feel like someone’s gonna take advantage of the policies or take advantage of a situation”. Creating a workplace culture that facilitates employee accountability in accessing CFWPs can be related to the sector and the nature of the work: “…It’s about people understanding what their accountabilities are and it’s up to them to figure out how they’re gonna fulfill those things… we’re a consulting company, right? So if you don’t do it, we can’t bill for it, which means we don’t have the money coming in…” (P4, F). The example of consultancy-based work illustrates the need for employees to remain accountable for their time in order for the business to thrive.

#### 3.4.4. Demographics

Populations globally are aging, fertility rates in Canada are declining, and more older adults are choosing to age in place given an overburdened healthcare system: “we want them where they can thrive and like they’re not really protected in the healthcare system right now because it’s overflowing” (P6, F). Workplaces that respond to the needs of evolving demographics place the organization in a better position to keep employees satisfied and healthy: “…it all depends on where the organization is in terms of demographics and the life cycle of those employees and their requests… It helps you guess what type of leaves and policies are really important…” (P5, F). Specific to caregivers in the workplace, P4 (F) noted that many workplaces are concerned with other demographics without a lens on CEs—who are primarily employed full-time, female and aged 45–65, representing the most skilled and experienced in the workforce [9]: “They’re already doing it for a variety of other demographics. This is another one, and the rationale still applies for why you want to be a more inclusive employer; ultimately, its attraction, retention, stronger engagement, all of that… this is another demographic that they need to work on and bring into their portfolio” (P4, F). Employers that are cognizant of their employment trends can better prepare for and assess the need to implement the Standard: “…because I still work with a pretty young group of people like I just happened to know that I’ve got maybe two right now who are in the process of potentially looking after parents, one is doing…a little bit of helping, but yeah, I’m really lucky that most of the others aren’t yet…” (P1, F). While presently there may not be a need for formal supports such as the Standard for CEs in the workplace, P1 acknowledges the evolution of the organization’s employees in recognizing the need for better supports for this cohort in the future.

### 3.5. Current Employee Supports

While most interview participants confirmed that the Standard is not currently implemented in the workplace, a variety of existing supports and accommodations for CEs were discussed, ranging from umbrella policies to emotional support. Most prominently, it was often noted that informal support on a case-by-case basis and work flexibility were offered under existing umbrella policies.

#### 3.5.1. Umbrella Policies

Only 24% of survey respondents reported the Standard was currently implemented in the workplace, with only one interview participant confirming implementation of the Standard. To implement the Standard, P9 (F) noted that “grief and bereavement leave are extended” while also acknowledging slight changes that allow for broader definitions of what is covered under existing policies: “taking away like sick days and calling them wellness days”. P9’s (F) organization offers 12 wellness days to all employees, including CEs, as well as an additional six days built in for additional caregiving needs. While the Standard provides a general guide to organizations for establishing CFWPs, it is easily adaptable: “I mean, we can’t go over the top, but every employee that works with us has to participate in short-term and long-term disability” (P9, F). P5 (F) addressed the ability to use existing programmes and policies as a framework for implementation of the Standard: “You have medical accommodations for people who are off for medical reasons, and then you bring them back, and gradually return to work. It’s the same format as far as accommodation… you customize it for returning to work… it’s the same” (P5, F). P7 (F) recognizes the potential for adoption of the Standard into existing practices as well: “… I think there’s a natural switch that can be made in just using terminology that would be more encompassing and wider reaching in its meaning…” (P7, F). This recognizes different ways of adopting the Standard; the Standard acts as a tool that facilitates the creation of CFWPs.

#### 3.5.2. Informal Support/Case-by-Case

Among most interview participants, it was observed that a common form of support offered to CEs in the workplace was on a case-by-case basis: “…our approach to our employee experience is a very non-HR one, not heavy administration. We still do a lot kinda case by case. We have some policies ‘cause obviously you have to have some” (P4, F). Further, P4 (F) posits that formal policies and practices are important, but the nuances across individual circumstances must be taken into account when determining what supports or accommodations to offer employees. Even within the realm of unions and collective agreements, the process of case-by-case assessment of employee needs was recognized as a necessary component of employee protection: “…but there will be individual circumstances …we may need to go beyond the collective agreement… so that you’re able to ensure the retention of the worker because often if you’re not able to have that accommodation, you’re forced to quit, and that is not a win–win situation for anyone…” (P4, F).

In contrast, P1 discussed the need for formal adoption of policy in the future due to the growing cohort of CEs amongst employees. P10 (F), the Director of Communications at a non-profit organization, also reflected on the general process of support and accommodation for CEs being offered on a case-by-case and often informal basis, despite the implementation of formal practices within the organization’s HR department. From the employee perspective, it was noted that case-by-case can be a desirable process for accessing supports or accommodations, recognizing that collaboration between managers and employees helps to shape an individual plan. Although it was acknowledged that this may not be the process that works for all CEs seeking support, formalized policies such as the Standard may lend themselves to accessing accommodation for fairness across different employee positions within the employee hierarchy.

#### 3.5.3. Flexibility

Offering the flexibility of working remote, hybrid, paid and/or unpaid leaves, and scheduling was observed to be a common CFWP offered by workplaces, as reported by P4(F), P5(F), P7(F), P9(F), P10(F) and P11(M). P4 (F) recognized how flexible working arrangements can shift and adapt to the evolving needs of the CE, in the interest of maintaining their productivity and engagement in the workplace. P4 also discussed flexibility already being in place from the onset of employees being hired. The nature of consultancy work allows for remote flexibility, making this an ideal scenario for a CE with changing caregiving responsibilities: “…we don’t have set hours, we don’t have set locations… so for folks who have care responsibilities, they can weave that into their workday, however is necessary” (P4, F). From the perspective of those without access to such accommodations, P2 (F) discussed the ability to step out of work in the afternoon to attend a caregiver support group, then returning to the workplace to make up for hours. Job sharing and capacity were acknowledged by P4 (F): “And we’ve also had people who have had to take, you know, extended periods of time, and we’ve found ways to figure that out and make it work”; and P10 (F): “We just picked up the slack. The others picked up where those couldn’t and we just, you know, we kind of muddled through”, as related to the ability to offer flexible work arrangements. An inclusive workplace culture that facilitates working as a team can allow for those requiring supports to lean on colleagues temporarily, a common tactic already employed at many workplaces within the context of sick days unrelated to caregiving. P10 illustrated an instance of an employee’s caregiving responsibilities temporarily impacting their ability to fully engage in workplace activities, allowing the employee to focus on work that they can “pick up and put down”, rather than diving into more complex projects.

The COVID-19 pandemic was discussed as facilitating a major shift in the workplace with regard to flexibility. P9 (F) reflected on the workplace shifting to fully remote work for all employees: “But during COVID early days, like we had one person who… was a caregiver full time at home with an elderly person. And there was always… the flexibility of, if you have extenuating circumstances, we allow some flexibility around your hours of work… if you need to take somebody to the doctor’s appointment, that’s fine. Just make up your time when you can”. CEs in the workplace were able to better manage caregiving duties in relation to balancing paid work, allowing them the flexibility to take their adult care dependents to appointments while making up for work hours outside the general schedule structure. This further exemplified the organization’s ability to be supportive of CEs moving forward, where P9 (F) acknowledged the functionality and sustainability of remote work within the organization and extended this beyond the restrictions of the pandemic.

#### 3.5.4. Emotional Support

Outside the context of accommodation, it is sometimes as simple as offering acknowledgment and emotional support to CEs, which mitigates perceived stress. P1 (F) discussed maintaining good rapport and open dialogue with employees through frequent phone calls with site staff, and further acknowledged how this has leveraged into employee loyalty and retention of talented staff: “They are comfortable calling me… to take like a half hour and debrief with me… we have valuable people who are now willing to stick with me, essentially through thick and thin because I’m fighting for them”. P1 (F) further recognized that offering emotional support to her employees sustains the psychological health and safety, as employees feel safe to express health concerns. P1 (F) acknowledges that often employees simply want to vent and are not necessarily looking for any tangible solutions or accommodations. In some cases, when emotional support is not extended to the employee and concerns are not acknowledged, resentment towards the employer can fester. P6 discussed that a lack of compassion and emotional understanding from the employer can result in avoidable turnover of staff.

#### 3.5.5. Measurement and Feedback

An integral component to the ongoing sustainability of CFWPs within an organization, regardless of whether the Standard is formally adopted or not, is measuring the value of current practices and gaining a sense of employee perspectives on the improvement of policies. P5 (F) discussed the ease of employing such strategies in today’s tech-laden work environments: “you don’t have to get a third party in to do all this…”, with the ability to easily create and distribute pulse surveys to measure the effectiveness of policies in meeting employee needs. P5 (F) further illustrated the value in collecting feedback to establish how familiar employees are with certain workplace policies available to them, while developing communication strategies to better inform employees of available supports. As a facilitator of employee experience, employee surveys can provide insight into employee demographics as well as employee needs: “…it’s not good enough to say, ‘well, I gave them the policy and they acknowledged it’” (P5, F).

For P9 (F), formal adoption of the Standard entails distributing “really in-depth staff satisfaction surveys”, specific to supports. While P4’s organization has not formally adopted the Standard into workplace policy, the concept of evaluation of employee experience relevant to caregiver supports takes place via regular employee check-ins and exit interviews. This type of evaluation leads to improvements, relevant to CFWPs: “…we think we’re doing a great job and then the staff will say ‘well, you know what we need this’, but we’ve also identified a lot more needs…at first it was being really family friendly…we keep having new elements introduced in a more formal way” (P10, F). Within the context of unions, the evolution of policies and collective agreements often relies on measuring what is currently working and not working, recognizing that demographic shifts across geographic locations of membership incite different needs: “…it’s gender based as well as looking at…different ethnic communities…and we also learn from our workplaces” (P7, F). Collecting employee feedback can inform policy evolution.

### 3.6. Barriers to Uptake

Given minimal uptake, a variety of factors were reported to hinder the formal adoption of the Standard. C-Suite disconnect, access to HR/capacity of HR, and spatiotemporal conflicts based on sector were overwhelmingly the most prominent barriers within the complexity of workplace structures. Further, simply understanding what formal adoption of the Standard entails appears to be a glaring hindrance, including navigation of the Standard document itself. Amongst other perceived barriers were an overall lack of policy within the workplace to guide the adoption of new practices.

#### 3.6.1. Spatiotemporal Constraints

The type of workplace sector and the nature of the work done at an organization were determined to directly impact an organization’s ability to adopt the Standard. P1 (F) manages emergency medical responders and paramedics across different worksites, recognizing many of these individuals are CEs who are simultaneously balancing their work in healthcare with their care responsibilities at home—known as Double-Duty Caregivers (DDCs). Site-based work often prevents P1 from being able to flexibly accommodate her DDCs with respect to balancing their work and care: “If somebody can’t show up that day it’s like… now what are we going to do? Cause other lives are now impacted… I’d 100% personally support them and it’s like now how can I”. Further, P1 (F) discussed the additional challenge of remote-based work for the first responders: “And it’s for a remote location, so it’s not like people could just, say, zip on in to take their [care recipient] to an appointment… I mean, it should have been addressed a while ago, but it’s been kind of like I’m stuck in corner”. Components of the Standard, such as implementing flexible work arrangements becomes challenging when the workplace structure requires employees to be in person, especially in remote areas: “…we have five rural sites too, so then you can add a little layer of complexity to rural… they may be 3 h away from this support group…” (P2, F). Personal Support Workers (PSWs) represent a subset of DDCs that experience spatiotemporal challenges in balancing work and care, where having an employee cover a shift so another employee can take a care dependent to an appointment is not always a simple solution: “…when you’re dealing with a dementia patient… it doesn’t lend itself to have a sub come in… those circumstances are a bit trickier” (P9, F). Fiscally, many organizations cannot easily support accommodations that allow for employees to leave the workplace for caregiving duties during the workday without having a “floating staff that helps in these circumstances” (P7, F). P11 (M) recognizes that given these challenges, many organizations are unaware of potential solutions to help mitigate these space-time conflicts: “They want to be able to do more for their employees, but they just don’t really know how, and especially with the larger kind of organizations that we do work with and some of our partners”.

#### 3.6.2. C-Suite Disconnect

A major barrier to the uptake of the Standard that was consistently reported was the notion of a perceived disconnect between the C-Suite and employees at a lower level. Rather than assessing the needs of current employees and developing solutions to ensure CEs receive the support needed, the C-Suite can often be preoccupied with compliance. Those at the top of any organization are obligated to ensure mandated employee rights and accommodations are being implemented within the workplace, often neglecting to do anymore that what is required by law: “…the compliance takes over the other stuff, and because caregiving isn’t really legislated yet it’s not a priority…” (P5, F). Unless employers are mandated to adopt the Standard into practice, it is up to the discretion of the workplace to determine whether this is a priority.

Managers on the ground may see the need for more tangible supports in the workplace; however, C-Suite executives of larger companies may not be close enough to assess the same need. P1 in the primary resource sector discussed anticipated conflicts around convincing higher-ups of the need to begin carving better policies and practices: “…they don’t see it the same way that we see, like when our people come to us super upset because this is all going on at home”. It was further discussed that this disconnect can be a result of lack of experience in caregiving, resulting in a lack of understanding for the need to support CEs: “…it’s the old boys club where they haven’t had to look after people, they have not been in these shoes, so they don’t have that understanding” (P1, F). Having a champion within the workplace that is represented amongst C-Suite is crucial for a top-down approach to effectively implement culture and policy change in the workplace: “…part of my aspiration was I just want to be the kind of leader that I wish that I had had during my years… when I was a caregiver… So, it’s very important to me to make people feel supported…” (P9, F); “…having the champions to talk about it and see where there are opportunities that arise… it’s part of a toolbox” (P7, F). In P1’s experience, a lack of awareness of employee needs amongst C-Suite can act as a significant roadblock to solutions for CEs: “but I know if somebody…can just work half time and I’m actually, honestly, perfectly fine with job sharing, I think it would be a great thing. But upper management is like, no, that’s a bad idea” (P1, F). In the case of P1 (F), the C-Suite acts as a barrier to potential solutions around accommodation for CEs: “They don’t have any ability for people to switch shifts and like…the payroll side of things. And I’m saying why not?” (P1, F). Those in upper management and C-Suite may also view certain CFWPs as requiring too many resources to implement: “… I think there is this perception, this hesitancy that it’s going to take a lot of resources, or it’s ‘do we need it right away’ or that kind of thing”, P5 (F).

#### 3.6.3. Access to HR

Human Resources (HR) is a catalyst for CFWPs across many sectors and workplaces of various sizes. HR is often the first point of contact for employees entering a new workplace, accompanied by employment packages and onboarding. Throughout an employee’s tenure at an organization, HR often also remains the channel for inquiring about employee benefits and clarification around policies. Whether it be a large HR department, an HR representative, or an employer of a small business responsible for employee experience, CEs are reliant on these open channels of communication for maintaining a healthy balance with work and care. For larger organizations with existing HR departments, the “operational strategic plans and targets” (P5, F) of the organization directly impact the employee experience. While the focus can often surround larger market trends, the needs of employees can be left unidentified and unmet: “organizations… they want to lead and make sure that their sustainable … they’re always looking at your markets, your products, your services. But there has to be an element of internal environment, and it shouldn’t always just be ‘HR will handle that’” (P5, F). The direction of HR initiatives is at the mercy of upper management based on organizational planning strategic directions, often with an emphasis on compliance and meeting market trends to remain competitive for recruitment: “…bigger organizations want to be on trend and want to be the leader … it all comes down to what’s urgent? What’s priority? What’s in our strategic plan that we need to focus on, the board’s after us for reporting and stuff…” (P5, F). P5 (F) further discussed the many challenges associated with HR initiatives not being included in the strategic plans of an organization. Initiatives such as the implementation of the Standard are then left to the wayside, without the support of upper management giving the green light to HR to drive the initiative forward.

Alternatively, smaller organizations with limited HR capacity may not have the manpower needed to support the development and implementation of HR initiatives that are intended to support CEs. P10 discussed these struggles: “… in terms of actually implementing the actual Standard, we haven’t done it…We’re small for 80 to 100 people. We have no resources… It’s great if you’re a huge department. We’re tiny. HR department has to dedicate a person to try to do this…”. P5 further recognizes the processual nature of such organizational changes, acknowledging the various working parts to sustainably implementing new workplace policies: “… it should also be embedded in the culture where the leadership believes this is the right thing to do…” (P5, F). A lack of capacity or access to HR can prevent prioritization of implementing the Standard. With regard to recruitment and retention, it was noted that supporting the implementation of employee protections such as the Standard can help to recruit and retain skilled staff, “…Why is everything reactive? It’s a good thing to have… always talking about the war on talent, and we want the best people… that will drive people to be part of this organization because you’re showing you care for your employees”, P5 (F).

#### 3.6.4. Standard Navigation

From the CSA download data of both the Standard and Handbook up to 31 March 2024, the Handbook has significantly more downloads than does the Standard. Respondents of the survey indicated the Handbook was found to be more useful than the Standard, with 61% reporting they either “agree” (37%) or “strongly agree” (24%) that the Standard was useful, and 65% reporting they either “agree” (34%) or “strongly agree” (31%) that the Handbook was useful. This suggests that the Handbook is a more user-friendly document for professionals to follow. P5 (F) discussed limitations with the Standard as potentially contributing to minimal uptake: “… as an HR professional trying to get this implemented, just tell me why. I know it’s good, and we should be doing it, but what do I need to do? How do I actually do it?… I think most organizations struggle with how do we do this” (P5, F). Within the context of workplace standards more generally, voluntary or involuntary, P10 (F) discussed the issue of document navigation: “… they get overwhelmed and then they do nothing” (P10, F). P5 discussed the issue of how to navigate the Standard and bring it into practice, also relates to who exactly is responsible for such an initiative: “Who’s going to do it? Who has time… what are we all responsible for?” (P5, F). In addition to challenges around navigating the Standard and its implementation, interpretations of what exactly formal adoption of the Standard entails remains ambiguous: “Because I think what the Standard feels like is something very rigid that you have to adopt, that you can’t cultivate or you can’t take apart piece by piece…” (P11, M). Misinterpretation of what formal adoption of the Standard entails, coupled with challenges around navigating the Standard document itself, and understanding how to apply it within the context of the organization’s capacity, all contribute to minimal uptake of the Standard. As it is voluntary, formal adoption takes different forms across organizations.

### 3.7. Facilitators to Uptake

Given the voluntary status of the Standard, widespread adoption of the Standard in workplaces across Canada is reliant on agents of influence. Outside the influence of an organization’s values and culture, formal adoption of the Standard was noted to be influenced by trends in Equity, Diversity, and Inclusion (EDI), often related to strategies of recruitment and retention. The potential for uptake of the Standard in Union Collective Agreements provides a channel of compliance that can mandate support for CEs in workplaces with employees protected by Union membership. P1 (F) discussed concerns around the prospect of implementing the Standard within the organization, being contingent on a government mandate that says to organizations, “you have to do this”. It is interesting to note that the top sectors offering CFWPs are healthcare and education [15], the most prominent sectors among survey respondents. P3 (F) recognizes this as potentially being more heavily influenced by the government: “… both of those sectors… are ultimately under the government, I think that speaks volumes…” (P3, F), illustrating that increased uptake of the Standard across workplaces in Canada is most effectively achieved through government mandate and compliance.

#### 3.7.1. Equity, Diversity, and Inclusion (EDI)

The issue of creating and implementing a workplace that supports CEs is an EDI issue: “…having equal opportunity for all regardless of your barriers that are not allowing women to be promoted, for example…” (P9, F). Nurturing an inclusive work environment and culture recognizes that without support and accommodation, CEs are disadvantaged in the workplace: “…it’s primarily women…when we’re talking about like gender equality…advancing women in the workplace…we need to acknowledge and kind of incorporate caregivers into that conversation” (P8, F). Recent literature highlights the gender disparities impacting CEs, with the majority of CEs in Canada being women, providing an average of three hours more of care per week than men [9]. Consequently, women accounted for around 60% of those who left the workforce due to caregiving in 2018; women were also more likely to reduce their hours and go on leave. Implementing supports for CEs, and by extension, women, in the workplace creates more equitable conditions for CEs to remain competitive participants in the workforce. P4 recognizes that formal implementation of the Standard can fall under the “inclusivity umbrella” within an organization and is a “really smart move”, with the benefits of doing so speaking for itself. P8 (F) noted a lens of gender equity shaping caregiving as an issue of EDI: “…we’re looking at gender equity, discrimination… incorporating caregiving into that conversation”. There are many working parts to EDI within an organization, where the workplace culture is informed by “the biases, the discrimination, the toxicity of an organization”, as noted by P5 in acknowledging the influence of workplace culture over EDI initiatives. EDI policies and initiatives are emerging more prominently across workplaces in efforts to reduce barriers experienced by individuals of different demographics, resulting in copious amounts of response strategies: “… the struggle organizations are having with inclusivity in general, right? Like, think of how many DEI policy standards are out there… there are organizations that have created their like you know, Chief DEI officer…” (P4, F).

#### 3.7.2. Unions

With the absence of a federal mandate, unions can act as an effective channel for regulating protections for CEs in the workplace. In cases where a CE’s caregiving responsibilities fluctuate, collective agreements can be a significant job protection strategy for members of a union:

As an example, we had one of our support staff whose mother was very ill, and she was sharing the caring duties with her sister, and she had used up her ten days. And then she had to use additional days to look after them. But in the course of those 10 days that she had, she developed, you know, illnesses herself, the psychological burden, and was not able to return to work, and so many have short-term disability programmes as well that are occupational, nonoccupational. You can rely on those as well. And then now she’s cycling back to the workplace. Three days for the next three weeks, and then… four days and then within a month, you’re back to five days a week. So, it does give you that sort of gradual return and then the time at home as well to look after yourself and still look after your family as well (P7, F).

P7 noted as a union representative that resistance to uptake of the Standard in collective agreements was determined to be a result of umbrella policies that capture the essence of what the Standard offers, without formal uptake of the policy: “… absences to take care of immediate family members… it may not be labelled as caregiver, but those are already in collective agreement” (P7, F). A standard specific to CEs in the workplace may be ranked lower on the priority list: “… getting more support on the Union side, that starts to get built in a bit more into those conversations… although I would say they probably would be like… we’ve got bigger fish to fry” (P4, F), and therefore contributes to minimal uptake of the Standard.

## 4. Discussion

The findings emphasize the crucial importance of establishing and sustaining an inclusive workplace culture while also acknowledging the current strategies and practices of CFWPs outside the formal adoption of the Standard, as well as the barriers organizations face in implementing the Standard formally. Future direction towards more widespread adoption of the Standard was thematically analyzed and established under the umbrella of EDI principles and practices, recognizing the need for both private and public policy for caregivers. With regard to whether there were existing accommodations for employees outside the implementation of the Standard, 55% of survey respondents responded with either “agree” ([*n* = 25], 35%) or “strongly agree” ([*n* = 14], 20%), with just 24% [*n* = 17] of survey respondents reporting that the Standard has been formally adopted in the workplace. This indicates that while workplaces are not formally adopting the Standard, there are other CFWPs that organizations are using to support CEs. This was further confirmed in the thematic analysis, where many participants confirmed access to supports and accommodation for employees with caregiving duties outside the adoption of the Standard.

Results of the logistic regression analysis indicate that organizations that have fully implemented all aspects of the Standard (OR = 15.8, *p* = 0.003) into their workplace will recognize the formal adoption of the Standard. Additionally, those with existing supports in place for CEs (OR = 15.9, *p* = 0.002), whether that be through existing policies or programmes, are more likely to recognize formal adoption of the Standard to formally recognize the existing supports. The thematic analysis indicated there were many circumstantial factors related to whether workplaces have adopted principles of the Standard (either formally or informally), including workplace size, capacity, sector, and culture. Misconception and confusion around what exactly constitutes the formal uptake of the Standard appear to be a significant driver of minimal uptake. The Standard is voluntary for organizations; it is then up to the organization to establish what works within the parameters of the workplace structure and current policies to represent formal adoption of the Standard. The Standard was designed to recognize and support the differences in workplace characteristics and size, allowing for the guidelines to be adapted within the context of the workplace and CE needs. The voluntary nature of the Standard means there is no accreditation tied to formal adoption of the Standard. Future research on the Standard may stand to benefit from measuring the extent to which those engaging with and those who are aware of the Standard maintain an understanding of what formal adoption of the Standard can entail for more accurate evaluation of uptake.

Workplace culture was consistently most commonly noted as a determinant of whether an organization was open to adoption of the Standard, either informally through CFWPs or through formal adoption of the Standard into the workplace policies. Additionally, the notion that the Standard could provide better guidance to HR professionals and employers looking to create a workplace culture that is more inclusive of CEs was acknowledged. It may not be immediately clear to an organization as to who is responsible for the implementation of the Standard, and exactly what steps are needed within the context of the organization to establish support for their CEs. The recently launched online course “Creating Caregiver-Friendly Workplaces,” noted above, offers a more user-friendly, step-by-step guide on how to put the Standard into practice, including case studies for more contextual learning. The course can support organizations that are unsure of how to approach the Standard by offering more guidance on who is responsible and what exactly needs to be performed. Within the frame of workplace culture, establishing an atmosphere of open communication between employers and employees without CEs feeling fear of repercussion towards their job safety is an important piece of fostering an inclusive workplace environment. Abuse of any workplace policy can be mitigated by an inclusive and supportive workplace culture, inciting feelings of loyalty and reciprocation amongst employees.

Increased awareness around the Standard as a voluntary workplace policy and clarification around what defines formal adoption are needed, as is increased awareness of current labour force demographics with respect to the increasing population of CEs and dynamic needs across the life cycle. Although over 60% of survey respondents indicated their workplace is aware of CEs, some interview participants discussed the hindrance that a negative workplace culture can have on upper management’s ability to recognize the need to better support and acknowledge CEs within the organization, inciting the vicious cycle of discouraging CEs from identifying in the workplace. Flexible work arrangements and customized support on a case-by-case basis were discussed to be the most prominent CFWPs. This, in a sense, represents informal adoption of the Standard under existing umbrella policies or emerging policies through expansion of definitions and scope of certain benefits. Workplaces that were able to maintain a positive and inclusive workplace culture for CEs emphasized the importance of measurement and feedback of employee experience, evaluating the effectiveness of current practices in meeting the needs of CEs. Raising awareness of the Standard as an EDI issue can help to facilitate a better understanding of what adoption of the Standard means for an organization. Integrating the Standard under the umbrella of EDI policies or initiatives within a workplace presents an opportunity for organizations to broaden the scope of inclusivity and directly contributes to facilitating culture change.

Several barriers to uptake were identified, pointing to potential improvements to the Standard that may help organizations mitigate these challenges. Specifically, the workplace sector can act as a barrier to uptake if the nature of work is typically site-based and requires physical attendance in certain locations. More creative solutions are needed to assist organizations in retaining staff through accommodation while also operating within the capacity of what the organization can justifiably offer. In conjunction, a lack of support or consideration from upper management and the C-Suite can create resistance to changes in workplace culture and policy if the need for such changes is not clearly recognized.

In comparison to other countries, such as Australia, Germany, and the United Kingdom (UK), Canada lags behind with respect to workplace and government supports for CEs [12]. According to a report published by the United States Government Accountability Office in 2020, Australia launched a National Carer Strategy back in 2009 to mitigate negative impacts on CEs’ labour force participation, which led to the Fair Work Act in 2009, legislating flexible work arrangements for CEs [54]. In the UK, the Children and Families Act of 2014 extended the right to request flexible work arrangements to CEs, making organizations compliant with offering CEs flexible work arrangements. In Germany, the Family Caregiver Leave Act, established in 2012, provided CEs in Germany the right to reduce their hours temporarily. Further, the report noted that the existence of these legislative acts in the noted countries provides CEs the right to access flexible work, which has been shown to improve job security and work–life balance for CEs [55].

### Limitations

The primary limitation that significantly impacted recruitment, data collection, and analysis of the present study surrounds the Canadian Anti-Spam Legislation (CASL) which restricted the CSA from contacting those who have downloaded the Standard and Handbook beyond the previous two years. The primary objective of connecting with all those who have engaged with the Standard and Handbook was negatively impacted by this restriction, resulting in the expansion of research objectives to include not only organizations directly engaging with the Standard but also those engaging with CFWPs more broadly. This study excluded the downloads of the French versions of the Standard and Handbook due to the low number of downloads. While the CSA’s standards are national in reach, Quebec refers to its own standards association, known as the Bureau de normalisation du Québec (BNQ). Quebec more readily recognizes the standards associated with the BNQ for their organizational standards, as opposed to the Standards associated with the CSA, potentially contributing to the low uptake of the CSA B701 French Standard and Handbook.

Further, given the CSA’s privacy policies around access to the contact list of those who have downloaded the Standard and Handbook, it is possible that some if not a majority of the downloads of the Standard were not applicable—i.e., individuals seeking to download the Standard for curiosity purposes, not tied to an organization or position of authority. It is also possible that some or many of the downloads are multiple downloads from the same individual. Given that the sample size was initially determined based on the total English downloads of the Standard and Handbook, issues around potential duplicates or ineligible respondents impacted the anticipated sample size, in addition to large impacts created by the CASL. As the CASL meant only those who had downloaded the Standard and Handbook within the previous two years were contacted for response, the sample of survey participants does not provide a representative sample of all the organizations that have downloaded the Standard and handbook, and more broadly, organizations across Canada with regard to size and sector. This limitation was unavoidable as the present study operated under the assumption that the majority of Standard and handbook downloads are by those in senior or executive positions directly representing an organization. As a result, the interview sample size of *n* = 30 was not reached. However, thematic analysis of interview data reached data saturation as no new themes began to emerge upon the 11th interview with the final participant, eliminating the need for a sample size of *n* = 30. A primary incentive of the selected methodology is to recruit participants from the survey who opt to be contacted for a follow-up interview. However, only three survey respondents opted to be contacted for a follow-up interview, and consequently, the remaining interview participants were recruited from the noted existing organization network, potentially creating some biases in the interview results. As recruitment was expanded to include an existing organizational network of over 30 organizations, potential biases and influence during data collection or interpretation surrounding researcher reflexivity present potential limitations of this work. The context of connection to this organizational network is within the realm of those actively engaged in or aware of CFWPs and therefore may present some biases in best practices of CFWPs or the best use of the Standard.

Another limitation of the study is the low response rate for the survey (*n* = 71), making the results (i.e., descriptive statistics and regression model) less generalizable to the Canadian landscape of organizations engaging with CFWPs. In addition, only one of the interview participants was male, which presents implications for the generalizability of the results, given that 52% of caregivers in Canada are women and 48% are men [9]. Provided that responses of the survey were not mandatory—resulting in blank responses across the variables—the imputation of missing data presents limitations for the validity of survey results.

## 5. Conclusions

Results have provided insight into the degree to which participating workplaces have engaged with the Standard, as well as the barriers and facilitators to formal adoption. Continued knowledge mobilization (KMb) of the Standard and Handbook will be framed around awareness of what constitutes formal adoption of the Standard. Results of this study will directly inform the creation of promotional tools in the form of magazine articles, a webinar series, a research brief, and social media campaigning. Unions will be targeted, as will establishing supports and accommodations for CEs as an EDI issue, broadening definitions of current policies. From analysis of both the survey and interview data collected, it was determined that minimal uptake of the Standard has occurred, with some engagement in CFWPs across workplaces. Only 24% of survey respondents indicated that the Standard is currently implemented, and one out of eleven interview participants confirmed the Standard is formally adopted, with 42% [*n* = 30] of survey respondents reporting existing supports or resources were available to support CEs in the workplace. Although, logistic regression modelling indicates that the existence of supports for CEs within the organization is a strong predictor of formally adopting the Standard into workplace policies, this is coupled with whether the organization has implemented all aspects of the Standard, rather than only those that best fit the organization. Thematic analysis of the interview data provides direction for future strategic initiatives towards increasing uptake of the Standard across Canadian workplaces, reducing strain on the healthcare system by supporting CEs’ ability to manage their unpaid care. The most prominent of these strategies surround updating and improving the Standard to be user-friendly for organizations and provide more guidance around varying conditions across workplace sizes and sectors, while also raising more targeted awareness around the CCFWP course, given its in-depth approach to navigating the Standard and Handbook.

Insight into the implementation of CFWPs outside the formal adoption of the Standard is a critical lens of this work, as it will allow a better understanding of the approaches workplaces are taking to support the growing cohort of CEs in Canada. Studies have shown the benefits to both employers and employees in supporting CEs in the workplace through reducing costs around lost productivity, absenteeism, presenteeism (an employee’s inability to concentrate at work), and reducing turnover [18,19,20,55]. The business case for why organizations should support CEs is well justified, indicating the need to illustrate the benefits of adopting the Standard voluntarily, giving the organization a competitive advantage for recruitment and retention. Policy advocacy towards federally mandating the Standard in workplaces across Canada will also be targeted in the interest of facilitating the widespread adoption of CFWPs.

## Figures and Tables

**Figure 1 ijerph-22-00907-f001:**
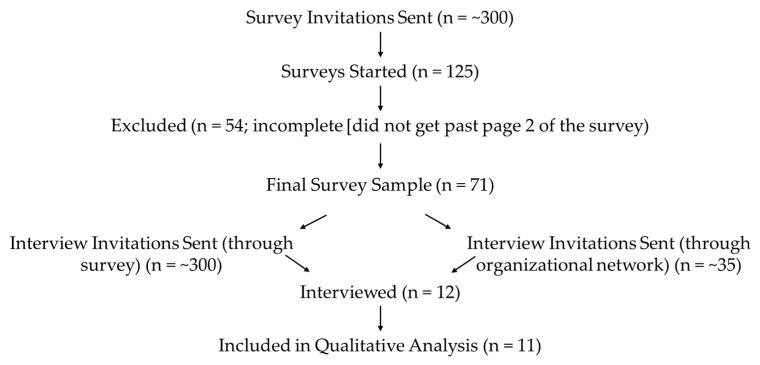
Participant flow diagram for the survey and interview phases of the study.

**Table 1 ijerph-22-00907-t001:** Descriptive statistics from survey analysis (*n* = 71), including frequency, percentage, and standard deviation.

Variable	Freq.	%	Std_Dev
What is your workplace sector?			
Healthcare	28	40	48.99
Education	19	27.14	44.47
Mining/Primary Resource	2	2.86	16.67
Transportation	2	2.86	16.67
Engineering	1	1.43	11.87
Manufacturing	1	1.43	11.87
Agriculture	1	1.43	11.87
Communications	1	1.43	11.87
Construction	1	1.43	11.87
Other	14	20	40
What is your workplace size?			
1–99	28	39.44	48.87
100–499	15	21.13	40.82
500+	28	39.44	48.87
How long have you worked at this organization?			
Less than 1 year	10	14.08	34.78
1 to 4 years	27	38.03	48.55
5+ years	34	47.89	49.96
How did you hear about the CSA B701 Carer-inclusive and accommodating organizations Standard?			
Email	19	26.76	44.27
Internet searches	12	16.9	37.48
Social Media	12	16.9	37.48
Newsletter	6	8.45	27.81
My organization	4	5.63	23.05
Word-of-mouth	4	5.63	23.05
Colleague(s)	2	2.82	16.55
Employee	1	1.41	11.79
Other	11	15.49	36.18
Does your organization have a Human Resources department and/or role?			
Yes	61	85.92	34.78
No	10	14.08	34.78
Is the Carer Standard currently implemented in your workplace?			
Yes	17	23.94	42.67
No	28	39.44	48.87
Don’t Know	26	36.62	48.18
Are there existing workplace supports and or resources for carer employees in your workplace?			
Yes	30	42.25	49.4
No	18	25.35	43.5
Don’t Know	23	32.39	46.8
Have carer employees identified themselves in your workplace?			
Yes	34	47.89	49.96
No	16	22.54	41.78
Don’t Know	21	29.58	45.64
Are you a caregiver and or have you been a caregiver in the past?			
Yes	55	77.46	41.78
No	16	22.54	41.78
If you answered yes to the previous question, indicate how many hours per week you spend caregiving			
10–20 h per week	14	19.72	39.79
Less than 10 h per week	16	22.54	41.78
More than 20 h per week	41	57.75	49.4
I and or my workplace has found the Standard useful			
Strongly Disagree	3	4.23	20.13
Disagree	4	5.63	23.05
Neither agree nor disagree	21	29.58	45.64
Agree	26	36.62	48.18
Strongly Agree	17	23.94	42.67
I and or my workplace has found the Handbook useful			
Strongly Disagree	0	0	0
Disagree	0	0	0
Neither agree nor disagree	24	33.8	47.3
Agree	24	33.8	47.3
Strongly Agree	23	32.39	46.8
I and or my workplace has found BOTH the Standard and Handbook useful			
Strongly Disagree	0	0	0
Disagree	0	0	0
Neither agree nor disagree	23	32.39	46.8
Agree	32	45.07	49.76
Strongly Agree	16	22.54	41.78
My workplace plans to implement the standard			
Strongly Disagree	2	2.82	16.55
Disagree	10	14.08	34.78
Neither agree nor disagree	19	26.76	44.27
Agree	29	40.85	49.16
Strongly Agree	11	15.49	36.18
My workplace has fully implemented the Standard			
Strongly Disagree	4	5.63	23.05
Disagree	21	29.58	45.64
Neither agree nor disagree	20	28.17	44.98
Agree	20	28.17	44.98
Strongly Agree	6	8.45	27.81
My workplace is aware of carer employees			
Strongly Disagree	6	8.45	27.81
Disagree	5	7.04	25.58
Neither agree nor disagree	11	15.49	36.18
Agree	29	40.85	49.16
Strongly Agree	20	28.17	44.98
My workplace promotes an inclusive workplace culture			
Strongly Disagree	1	1.41	11.79
Disagree	9	12.68	33.27
Neither agree nor disagree	13	18.31	38.67
Agree	32	45.07	49.76
Strongly Agree	16	22.54	41.78
My workplace provides accommodation for carer employees			
Strongly Disagree	9	12.68	33.27
Disagree	13	18.31	38.67
Neither agree nor disagree	10	14.08	34.78
Agree	25	35.21	47.76
Strongly Agree	14	19.72	39.79
My workplace values work–life balance			
Strongly Disagree	1	1.41	11.79
Disagree	13	18.31	38.67
Neither agree nor disagree	11	15.49	36.18
Agree	28	39.44	48.87
Strongly Agree	18	25.35	43.5
My workplace offers flexible work arrangements			
Strongly Disagree	7	9.86	29.81
Disagree	9	12.68	33.27
Neither agree nor disagree	11	15.49	36.18
Agree	28	39.44	48.87
Strongly Agree	16	22.54	41.78
My workplace actively seeks out ways to better support its carer employees			
Strongly Disagree	5	7.04	25.58
Disagree	21	29.58	45.64
Neither agree nor disagree	15	21.13	40.82
Agree	24	33.8	47.3
Strongly Agree	6	8.45	27.81
Carer employees feel comfortable identifying themselves in my workplace			
Strongly Disagree	8	11.27	31.62
Disagree	13	18.31	38.67
Neither agree nor disagree	11	15.49	36.18
Agree	26	36.62	48.18
Strongly Agree	13	18.31	38.67
Carer employees feel supported in my organization			
Strongly Disagree	7	9.86	29.81
Disagree	9	12.68	33.27
Neither agree nor disagree	24	33.8	47.3
Agree	20	28.17	44.98
Strongly Agree	11	15.49	36.18

**Table 2 ijerph-22-00907-t002:** Logistic regression modelling for predictors of workplaces adopting the Standard.

Predictor Variable	Odds Ratio (OR)	95% CI (Lower)	95% CI (Upper)	*p*-Value
Standard fully implemented in the workplace	15.8	2.76	90.4	0.0027
Workplace values work–life balance	4.67	0.66	33.2	0.108
Workplace has supports for CEs	15.9	3.99	63.6	0.0019
Provides care >20 h per week	0.25	0.04	1.53	0.108

**Table 3 ijerph-22-00907-t003:** The macro and micro themes resulting from the thematic analysis of interview data.

Macro Theme (*n* = 4)	Micro Theme (*n* = 14)
Workplace Culture	Communication
	Self-Identification
	Employee Accountability
	Demographics
2. Current Employee Supports	Umbrella Policies
	Informal Support/Case-by-Case
	Flexibility
	Emotional Support
	Measurement & Feedback
3. Barriers to Uptake	Spatiotemporal Constraints
	C-Suite Disconnect
	Access to HR
	Standard Navigation
4. Facilitators to Uptake	Equity, Diversity, and Inclusion (EDI)
	Unions

**Table 4 ijerph-22-00907-t004:** Interview participant (*n* = 11) sociodemographic characteristics.

Participant	Sector	Organization Type	Gender	Location
P1	Primary Resource	Emergency Medical Services	F	Canada
P2	Education	ECE	F	Canada
P3	Healthcare	Non-profit (provincial)	F	Canada
P4	Business	Consulting	F	Canada
P5	Business	Human Resources	F	Canada
P6	Healthcare	Research	F	Canada
P7	Business	Union	F	Canada
P8	Healthcare	Public Health	F	Canada
P9	Non-profit	Non-profit (Federal)	F	Canada
P10	Communications	Non-profit (Federal)	F	Canada
P11	Healthcare	Non-profit (provincial)	M	Canada

## Data Availability

The data presented in this study are available upon request from the corresponding author due to privacy and confidentiality concerns.

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
