# Peer review of "Evaluating the Uptake of the Canadian Standards Association (CSA) B701:17 (R2021) Carer-Inclusive and Accommodating Organizations Standard Across Canada"

_ijerph, 2025, doi:10.3390/ijerph22060907_

Round 1

Reviewer 1 Report

Comments and Suggestions for Authors

This manuscript aims to analyse the implementation status of the CSA B701:17 (R2021) in Canada. The authors use survey and interview data to identify factors related to the uptake of the carer-inclusive and accommodating organizations standard. Results indicate that 24% of employers utilizes the Standard into their workplaces and the workplace culture is a crucial determinant of adoption.

I have read the paper with interest. Although the study reveals some insights and empirical evidence on the subject matter, I have serious concerns on the manuscript, which are listed below.

  • The manuscript should present a clear diagram to inform the reader on selection of survey participants, response rates in each stage, dropouts, etc. for both survey and interviews.
  • Who are the survey participants? (Age, gender, other characteristics?)
  • There are many empirical limitations of the manuscript. The operationalization of variables and regression models should be revised. See below comments.
  • The manuscript should provide clear descriptions of measurement details and operationalizations of variables included in the regression analysis. What are the ranges of variables? A table of descriptions and additional descriptive stats (with means, standard deviation) would be helpful.
  • Are all the dependent variables binary? If yes, standard OLS model may not be the best fit.
  • Are the results robust to use of alternative model types?
  • Regression results miss important information such as number of observations.
  • No regression diagnosis is reported. Are standard errors robust? Is there any multicollinearity?
  • There are no justifications for choice of number of variables in regression models. Why does each model have different number of explanatory variables?
  • There are potentially omitted variables in most regression models.
  • Findings of Table 2 are not discussed in detail within the text.
  • Limitations of data analysis should be acknowledged.

Reviewer 2 Report

Comments and Suggestions for Authors

This manuscript presents a timely and important investigation into the uptake of the National Standard for Psychological Health and Safety across Canadian workplaces. The findings clearly indicate a minimal formal adoption of the Standard among the 226 organizations surveyed. The authors employ a mixed methods approach that significantly enhances the study’s rigor and depth—survey data provide a valuable overview of the adoption landscape, while qualitative interviews offer rich, contextual insights into the barriers and facilitators affecting implementation.

The study is underpinned by a robust methodological and theoretical framework, incorporating descriptive statistics, regression analysis, and thematic analysis, allowing for a thorough triangulation of data. This not only strengthens the validity of the findings but also enhances their generalizability and relevance beyond the Canadian context.

This research is both original and highly relevant to the fields of occupational health, social policy, and organizational management. It addresses a specific and timely gap concerning Carer-Employees (CEs)—individuals who balance paid employment with unpaid caregiving responsibilities. Although the CSA B701:17 (R2021) Standard was developed to support this group, the manuscript highlights the minimal uptake of this resource despite the significant number of Canadians affected (over 5.2 million).

The study makes a valuable contribution to the field by providing real-world, empirical evidence on the limited implementation of this caregiver support standard and by identifying specific barriers to its uptake—particularly related to workplace culture and inclusivity. The use of a mixed-methods approach, incorporating survey data (n=71) and semi-structured interviews (n=11), allows for both quantitative breadth and qualitative depth. This strengthens the study’s ability to explore both the scale of the issue and the lived experiences behind it.

Notably, unlike broader studies that focus on work-life balance or general employee well-being, this research hones in on a concrete policy initiative, assessing its practical implementation across Canadian workplaces. In doing so, it fills a gap in understanding how formal standards for carer inclusion are—or are not—being adopted.

That said, a few minor revisions would enhance the overall clarity and depth of the manuscript:

  1. Comparative Context: The originality and applicability of the findings could be further enriched by situating the Canadian experience within a broader global context. Comparing this standard to similar initiatives in other countries or organizations would deepen the analysis and broaden the study’s relevance.
  2. Clarification Needed:
    • The manuscript notes that only 173 French versions of the Standard and Handbook were downloaded and that these were excluded from the study sample. However, this raises questions about how the authors account for such low engagement in a bilingual country like Canada. Further reflection or hypothesis on this issue would be beneficial.
    • It is unclear how the sub-sample of participants was selected. The manuscript would benefit from specifying the inclusion and exclusion criteria, as well as any recruitment challenges faced.
  3. Main Research Question: While the aim is implied, the main research question should be explicitly stated to anchor the study more clearly for readers.
  4. Ethical Considerations: The paper should briefly mention the ethical protocols followed in the research process, including informed consent, and confidentiality. We understand McMaster University granted ethical approval.
  5. Reflexivity: Including a few sentences on researcher reflexivity—particularly regarding potential biases, positionality, or influence during data collection and interpretation—would enhance the qualitative rigour.
  6. Qualitative Rigor: The inclusion of more direct quotations from interview participants and their contextualization would further ground the thematic findings in participants' voices and enhance the reader's engagement with the material. That would give the study more ‘ecological validity’

Methodology:
The mixed-methods design is sound and well-suited to the research aims. The quantitative analysis (descriptive statistics and regression) provides insight into uptake patterns, while the qualitative analysis (thematic analysis of interviews) uncovers barriers and organizational factors. This dual approach enhances the validity of the findings.

Conclusions:
The conclusions are consistent with the evidence and arguments presented. They directly address the central issue of low uptake of the CSA B701:17 Standard and identify workplace culture and inclusivity as influential factors—aligning well with the study’s objectives.

Overall, the manuscript is well-structured, methodologically sound, and contributes meaningfully to the field of workplace mental health policy and practice. In its present form, the manuscript is publishable and I recommend it for acceptance.

Reviewer 3 Report

Comments and Suggestions for Authors

Thank you for the opportunity to review this manuscript.  Overall, this is an interesting and timely study, and the mixed-methods approach is a significant strength of the project.  However, I have outlined some current suggestions that I believe, if addressed, will strengthen the potential impact of the manuscript.

  • The authors state that “over half of the participants were recruited having not completed the survey.” Please provide more clarity on how these individuals were identified and approached, and potential impacts of the mixed recruiting methods.
  • With the relatively small sample size of 71, and regression modeling being used, it is important to disclose the results of your power analysis, and whether or not it indicates that your sample size is adequately sized for detecting meaningful effect sizes. Please provide details.
  • Please explore within the limitations the implications of your only having one male within your qualitative sample, and the impact of this on the generalizability of your findings.
  • It is common practice within qualitative research to include some participant demographics after each quote, so that readers do not have to go back and forth to your participant demographics table in order to understand the context of who is saying each quote.  I recommend that you add at least age and gender to each quote.
  • Please add participant age within the demographics table.

Thank you for the opportunity to review this interesting and meaningful work. I look forward to its eventual publication.
